# Regulation of NKG2D Stress Ligands and Its Relevance in Cancer Progression

**DOI:** 10.3390/cancers14092339

**Published:** 2022-05-09

**Authors:** Amber B. Jones, Abbey Rocco, Lawrence S. Lamb, Gregory K. Friedman, Anita B. Hjelmeland

**Affiliations:** 1Department of Cell, Developmental and Integrative Biology, University of Alabama at Birmingham, Birmingham, AL 35233, USA; amberj96@uab.edu; 2Department of Pediatrics, Division of Pediatric Hematology and Oncology, University of Alabama at Birmingham, Birmingham, AL 35233, USA; amrocco@uabmc.edu (A.R.); gfriedman@uabmc.edu (G.K.F.); 3IN8bio Inc., New York, NY 10016, USA; larry@in8bio.com

**Keywords:** cancer, stress, NKG2D ligands, immunotherapy

## Abstract

**Simple Summary:**

Immune surveillance during periods of cellular stress is necessary to maintain homeostasis. In distressed cells, the expression of stress (natural killer group 2 member D (NKG2D)) ligands is induced on cells to promote recognition by cytotoxic immune populations. This process is exemplified during cancer initiation, progression, and maintenance. Unfortunately, cancer cell adaptation yields multiple mechanisms to evade immune cell recognition. Therefore, extensive efforts have been investigated to induce stress ligands on tumor cells to complement immunotherapy efforts. In this review, we provide updates on the current regulatory mechanisms involved with both stress ligand induction and repression and offer areas of consideration as research on this topic progresses.

**Abstract:**

Under cellular distress, multiple facets of normal homeostatic signaling are altered or disrupted. In the context of the immune landscape, external and internal stressors normally promote the expression of natural killer group 2 member D (NKG2D) ligands that allow for the targeted recognition and killing of cells by NKG2D receptor-bearing effector populations. The presence or absence of NKG2D ligands can heavily influence disease progression and impact the accessibility of immunotherapy options. In cancer, tumor cells are known to have distinct regulatory mechanisms for NKG2D ligands that are directly associated with tumor progression and maintenance. Therefore, understanding the regulation of NKG2D ligands in cancer will allow for targeted therapeutic endeavors aimed at exploiting the stress response pathway. In this review, we summarize the current understanding of regulatory mechanisms controlling the induction and repression of NKG2D ligands in cancer. Additionally, we highlight current therapeutic endeavors targeting NKG2D ligand expression and offer our perspective on considerations to further enhance the field of NKG2D ligand biology.

## 1. Introduction

To maintain normal homeostatic conditions, immune surveillance in the body is necessary to ensure the clearance of damaged, infected, or neoplastically transformed cells [1,2,3,4]. However, dysregulated immune surveillance can strongly contribute to disease etiology and progression [5,6,7]. Specifically in cancer, there are a host of mechanisms that these cells implement to evade immune recognition and ensure sustained survival [8,9,10,11]. Many of these adaptive mechanisms are implemented to mitigate both intrinsic (excessive proliferation, oxidative stress, etc.) and extrinsic cellular (chemo-, radiotherapy, etc.) stress, which can negatively impact immune cell activity [12,13,14,15,16,17]. Adverse immune reactions in response to stress are evident in the natural killer group 2 member D (NKG2D) stress response pathway. After exposure to stress-inducing stimuli, there is a consequent induction of NKG2D ligands on target cells. NKG2D ligands are then recognized by NKG2D receptors on subsets of neighboring cytotoxic immune cells [18,19,20]. However, in many cancers, there is a dysregulation of the stress response pathway resulting in decreased levels of NKG2D ligands, which interferes with immune recognition by effector cell populations [21,22,23]. Mechanisms of NKG2D pathway regulation have been investigated by multiple groups, and much emphasis has been placed on the activity of NKG2D receptor bearing immune cell populations [19,24,25,26,27]. However, the mechanisms through which the NKG2D ligands are regulated have not been fully elucidated, and further investigation is warranted to provide additional molecular insight into this clinically relevant immune pathway. In this review, we examine the current understanding of the regulatory mechanisms that govern NKG2D ligands in cancer, and we provide insight into therapeutic modulation of the stress response pathway for anti-cancer treatments.

## 2. NKG2D Pathway Overview

### 2.1. NKG2D Receptor

As a master regulator of immune surveillance, NKG2D receptor activation is essential for downstream signal transduction by cytotoxic effector cell populations (Figure 1) [28,29]. The NKG2D receptor, encoded by Killer Cell Lectin Like Receptor K1 (*KLRK1*), is found on natural killer (NK) cells, natural killer T (NKT) cells, subsets of gamma delta T cells, and small populations of CD4^+^ T cells [30,31,32,33,34]. Distinct polymorphisms of the NKG2D receptor have been described and have varying roles within disease etiology [25,35]. Polymorphisms in the receptor have been shown to correlate with susceptibility of chronic hepatitis B [36] and protection against systemic lupus erythematosus [37]. In cancer, a guanine to alanine mutation (Thr72Alanine) displayed increased expression in control samples in a cervical cancer study, suggesting a protective effect against cancer progression through the modulation of NK and T cell activity [38]. Functionally, the transmembrane homo-dimeric protein structure of the NKG2D receptor allows for signal transduction via the utilization of adapter molecules, as the intracellular domains of this receptor lack signaling capacity [39,40,41]. Humans are thought to exclusively utilize the DNAX activation protein 10 (DAP10) adaptor molecule, as there has only been one identified isoform of NKG2D in humans [42,43]. However, alternative splicing of the *KLRK1* gene in mice results in two distinct isoforms of the NKG2D receptor, which warrants the utilization of two adaptor molecules, DAP10 and DNAX activation protein 12 (DAP12) [44,45]. Upon activation in humans, DAP10 recruits intracellular signal transducers, phosphatidylinositol-3-kinase (PI3K) and growth factor receptor-bound protein 2 (Grb2), to sustain the proliferation and survival of NK cells and to potentiate their cytotoxic effect [46,47]. Activation of the NKG2D receptor is induced by external changes to neighboring cells that promote signaling cascades that cause either an inhibition of homeostatic maintenance and/or induction of cellular stress (Figure 1) [19,48]. NKG2D receptor activity is further enhanced through additional immune-modulating stimuli, such as exposure to pro-inflammatory cytokines interleukin (IL)-15 and IL-2 [49,50,51,52]. Specifically, IL-15 promotes the phosphorylation of the DAP10 adaptor molecule to further support downstream signaling via the NKG2D receptor [53]. In contrast to NK receptors aimed at recognizing major histocompatibility complex (MHC) I molecules (HLA-A, HLA-B, HLA-C), the NKG2D receptor preferentially recognizes non-MHC I molecules known as NKG2D ligands [54,55,56].

### 2.2. NKG2D Ligands

There are a variety of NKG2D ligands, and the regulatory mechanisms that control their expression vary contextually [57,58,59,60]. NKG2D ligands have been identified only in placental and marsupial mammals [61]. Humans express two distinct categories of NKG2D ligands with a large degree of polymorphism that have developed over time, possibly in response to various microbial evasions [62]. One family consists of MHC Class I Polypeptide-related sequence A and B (MICA and MICB, respectively), while the other family of NKG2D ligands consists of a six-member glycoprotein family of UL16-binding proteins (ULBP1–6) [34,63,64,65,66]. Compared to humans, NKG2D ligands in mice consist of retinoic acid early inducible-1 (RAE-1) with five distinct isoforms (alpha-epsilon), three isoforms of histocompatibility H60 (a, b, c), and one homologous murine UL-16 binding protein (MULT-1) [67,68,69,70,71]. Further contributing to the diversity of NKG2D ligands, there is a high degree of polymorphic expression [24,72]. For example, there are over 60 distinct allelic sequence of MICA and 20 of MICB, and this, coupled with differential receptor binding affinity, can significantly impact immune effector-mediated cytotoxicity [73,74,75,76,77,78]. The varying binding affinity of the respective ligands to the NKG2D receptor is particularly high for immune reactions in terms of dissociation constant (K_D_) ranges [68,79,80,81]. In addition to the canonical NKG2D ligands, novel non-canonical ligands have been discovered that activate the NKG2D receptor (i.e., LETAL, MUTS) [82,83]. While it is important to recognize there are alternative molecules that can potentiate this pathway, this review will focus primarily on the foundational NKG2D ligands and their regulation by distinct stimuli with known roles in cancer development and progression.

### 2.3. Cell Type Specific Expression

As the NKG2D pathway can be modulated to advance anti-cancer therapies, there is a need to understand which cell populations are more resistant or susceptible to its targeting. The expression of NKG2DLs in healthy human tissues is generally low or restricted [63]: however, there can be the heterogeneous expression of each ligand based on its specific role, tissue type, regulation, and exposure to certain inducers [63,84]. For example, ULBP1 is found in human B cells and monocytes and may aid in hematopoiesis [85], whereas MICA, ULBP1, and ULBP3 are present in human bone marrow and early progenitor cells [86]. In addition to varying baseline expression, the ligands have differing responses to various stressors. For example, in hematopoietic tissues, stimulation with alloantigen or superantigen upregulates NKG2DLs in T cells [87], and myelomonocytic differentiation leads to an increase in ULBP1 [85]. Stimulation with various growth factors induces ULBPs in CD14+ monocytes, and both IFN-α and IL-15 can upregulate MICA expression on mature dendritic cells [88]. In gut epithelial cells, MICA may be present and increased in response to changes or disruptions in gut flora [63]. Lastly, in airway epithelial cells, the upregulation of surface NKG2DLs can occur in response to oxidative stress [89].

Off-target effects of any therapy can limit clinical implementation; therefore, identifying potential risks to the healthy surrounding tissue is critical [90,91]. Although studies demonstrated NKG2D ligand expression on normal tissue, other reports have emphasized the semi-restricted tumor-targeting potential of effector populations. For example, a 2015 study by Pereboeva et al. showed that stress-inducing stimuli, such as irradiation and chemotherapy, did indeed induce both MICA and ULBP2 ligands on both glioblastoma tumor cells and human astrocytes [92]. However, despite these ligands being expressed on the astrocytes, these cells were significantly spared from the cytotoxic effect of ex vivo expanded gamma-delta T cells in comparison to their tumor cell counterparts [92]. Additionally, work from Lamb et al. utilized normal human astrocytes as an experimental control for the cytotoxic potential of temozolomide-resistant gamma delta T cells toward glioblastoma cells, and no toxicity to the astrocytes was reported [93]. Importantly, these findings provide preclinical evidence supporting the safe implementation of NKG2D-based therapies without risk to the surrounding non-tumor tissue.

In addition to understanding the impact that NKG2D ligands have on normal tissue in response to cancer therapy, it is also important to acknowledge that NKG2D ligands do play homeostatic roles. The expression of NKG2D ligands serves as a vital signal to the immune surveillance system where NKG2D receptor-bearing cells promote the clearance of damaged tissue. For instance, in response to events such chronic inflammation, burn injury, and obesity, NKG2D ligands are induced: this results in the recruitment of effector cells to rid the affected area of damaged cells to maintain tissue health [94,95,96]. Since the NKG2D ligands are stress responsive, a process such as tumorigenesis can have an overlap of stimuli that regulate NKG2D ligand expression. For example, infection with the bacteria *Helicobacter pylori* (*H. pylori*) is an established risk factor associated with the onset of gastric adenocarcinoma [97]. A recent study by Hernandez et al. demonstrated that exposure to *H. pylori* induced MICA and ULBP4 in gastric adenocarcinoma cell lines, which promoted NK cell cytotoxic activation [97]. In addition, stress to the endoplasmic reticulum induced MULT-1 expression in human endothelial cells is also linked to tumorigenesis [98]. A functionally intact NKG2D system also had a protective effect against tumor initiation as evident in both a fibrosarcoma and ovarian cancer model, respectively [99,100]. With the relatively harsh conditions that tumors maintain, the exploitation of NKG2D ligands to enhance immunotherapies is a promising approach.

Another important cell-specific consideration for therapeutic intervention is the ability of certain malignant cell populations to evade this pathway. For example, chemo- and radiotherapy-resistant tumor-initiating cell or cancer stem cell (CSC) populations may have relatively lower levels of NKG2D ligands [101,102,103]. For example, acute myeloid leukemia (AML) stem cells evaded killing by NK cells through increased expression of the DNA repair enzyme poly-ADP-ribose-polymerase 1 (PARP1), which caused a repression of NKG2D ligands [104]. Similarly, stem cells in solid tumors such as glioblastoma have also been shown to evade immune system detection via dysregulation of NKG2D ligand expression. Zhang et al. determined that epigenetic silencing resulted in ULBP1 and ULBP3 ligand repression in glioblastoma [105]. Following treatment with the hypomethylating agent decitabine, ligand repression was reversed, and NK-mediated killing increased [105]. However, other reports using glioblastoma cells have noted the increased expression of stress ligands in cancer stem cell populations, supporting a heterogeneous expression pattern that is not yet fully understood [106,107]. Substantial efforts have been dedicated to manipulating and targeting CSC populations to make them more susceptible to cytotoxic killing by NKG2D receptor-bearing effector populations [108,109,110]. However, it is important to note that other cell populations critical for tumor biology also express NKG2D ligands: this pathway can influence the more differentiated, non-CSC tumor cells as well as macrophages, dendritic cells, B-cells, and T-cells [69,111,112]. Despite the expression of NKG2D ligands being necessary for immunoreactivity and immune surveillance, sustained expression can be detrimental to pathway activity [113]. Constitutive MICA and Rae-1ε expression in a cutaneous carcinoma model was associated with the downregulation of NKG2D receptor expression and diminished NK-mediated cytotoxicity via receptor internalization and lysosomal-mediated degradation [114,115]. Adding to the complexity of NKG2D pathway regulation, there are also heterogeneous induction mechanisms for stress ligand expression that are relevant for cancer etiology.

## 3. Positive Regulators of NKG2D Ligand Expression

The regulation and activation of the NKG2D receptor has been extensively reviewed (see Refs. [19,24,25,26,27]); therefore, we have focused on the cancer-relevant regulation of NKG2D ligands and the potential for developing novel, targeted cancer therapeutics related to this pathway.

### 3.1. DNA Damage

Many current cancer treatment strategies rely on initiating DNA damage in malignant cells: DNA damage has an important role in immunoreactivity, specifically as it pertains to NKG2D ligand expression [116]. In transduced ovarian cancer cells, DNA damage from ionizing radiation and certain chemotherapies induced both human (MICA/B, ULBP1-3) and mouse (RAE-1) NKG2D ligands [117]. Along with ligand induction, there was increased activity of DNA damage response (DDR) pathway regulators, such as ataxia telangiectasia mutated (ATM) and ataxia telangiectasia and RAD3 related (ATR): their activity was supported by an increased phosphorylation of downstream pathway mediators Chk1 and Chk2 and the tumor-suppressor protein p53 [117]. Other cancer types, such as multiple myeloma and leukemia, also have a dependency on ATM and ATR for NKG2D ligand induction [118,119].

Along with the direct influence that the DNA damage response has on NKG2D ligand expression, there is also evidence that indirect mechanisms can contribute to ligand induction [120]. For example, as DNA damage occurs, there is an increased presence of cytosolic DNA that activates corresponding sensory pathways. These pathways can further activate the stimulator of interferon (IFN) genes (STING) pathway, which is an established immune regulatory pathway [121]. Lam et al. determined that RAE-1 expression in lymphoma cell lines was dependent on the activation of IFN regulatory factor-3 (IRF3), whose phosphorylation was regulated by DNA sensor pathway activation [122].

In addition to cytosolic sensory pathways, downstream molecular targets of DNA damage, such as the transcription factor and tumor suppressor protein p53, have also been shown to influence the expression of NKG2D ligands [122]. In lung cancer models, p53 mutational status influenced the expression of ULBP1 and ULBP2, as these ligands are direct transcriptional targets of p53 [123]. Cancer cells transduced with wild-type p53 resulted in a significant upregulation of the ULBP1 and ULBP2 ligands as compared to those with mutant p53 [123]. In the context of DNA damage, p53 is a downstream target of activated response elements, which promotes ligand expression, suggesting its necessity [117,124,125].

### 3.2. Cell Cycle Alterations

While DNA damage is one of the more established mechanisms that regulates NKG2D ligand expression, other phenotypes associated with cancer initiation and maintenance have also been implicated in ligand regulation. Malignant cells commonly have alterations in their cell cycle, and investigators are working to elucidate mechanistic links between cell cycle regulators and immune surveillance. An established family of cell cycle regulating transcription factors, the E2 factor family (E2F), is a direct regulator of NKG2D ligands. Specifically, the Rae family of NKG2D ligands are direct transcriptional targets of E2F, and increased overall E2F family (1,2,3) transcriptional activity leads to increased proliferation and Rae family expression [126]. E2F is also a direct phosphorylation target of ATM and ATR kinase activity. Therefore, E2F provides further regulation of NKG2D ligands by the DNA damage response pathway.

In addition to pathway regulators, the behavior of malignant cells has been linked to the expression of NKG2D ligands. The induction of hyperploidy in various cancer cell lines by cytochalasin D, a microfilament polymerization inhibitor, resulted in a subsequent upregulation of MICA and an increased NK cell cytotoxic response [127]. In multiple myeloma cells, a high-self renewal capacity resulted in increased NKG2D ligand expression and preferential targeting by NK cells [128]. As energetic output is often associated with increased cell division, some studies have determined that nutrient availability influences ligand expression. For example, enhanced mitochondrial citrate metabolism, glycolysis, and fatty acid metabolism promoted either ligand expression (MICA/B) or enhanced NK cell functionality in their respective tumor models, which supports the potential targeting of metabolically active cell populations [129,130,131,132].

### 3.3. Oncogenic Transformation

The acquired genetic changes during cancer can also influence NKG2D ligand expression. During oncogenesis, there is an immune landscape shift that contributes to disease progression [133,134,135]. Genetic alterations commonly associated with tumor formation have been reported to regulate NKG2D ligand expression. Unni et al. demonstrated a novel link between NKG2D ligands and genetic transformation: the combination of increased oncogenic Myc activity and diminished tumor-suppressor activity of either p53 or Arf resulted in an increased surface expression of Rae-1ε during lymphomagenesis [136]. Furthermore, Schuster et al. determined that not only did p53 need to be lost but the anti-apoptotic protein BCL2 also needed to be overexpressed to induce expression of the murine NKG2D ligand, MULT-1, in a Eµ-myc model of lymphoma [137]. These multiple oncogenic “hits” were necessary to facilitate NK cell-mediated killing of the lymphoma cells. An important distinction to note is that diminished tumor suppressor activity of p53 can regulate ligand expression independent of the DNA damage response cascade.

The phosphatidylinositol-3-kinase (PI3K) pathway is also commonly altered during transformation and implicated in NKG2D signaling. In a model of cellular stress in mouse fibroblast infected with cytomegalovirus, the PI3K catalytic subunit, p110𝛼, induced RAE-1 family ligands. Inhibition of this subunit caused a reduction in expression of RAE-1 ligands [138]. PI3K-mediated regulation of ligands has also been demonstrated in human breast cancer cell lines. Specifically, the dimerization of epidermal growth factor family members, HER2 and HER3, upregulated MICA and MICB expression via the PI3K axis, which increased NK cell recognition and killing of tumor cells [139].

Upstream of both Myc and PI3K, the oncogene RAS also influences stress ligand expression. Constitutive activation of mutant RAS in a human breast cancer model resulted in an increase in downstream pathway mediators such as mitogen-activated protein kinase (MAPK) and PI3K and in murine RAET1 ligand expression in mouse breast cancer cell lines [140]. Although established as a regulator of stress ligand expression, there was no dependency on enhanced ATM or ATR expression to substantiate NKG2D ligand expression in this model. This supports the heterogeneous regulation of NKG2D ligand expression independent of established pathways, such as that of the DNA damage response pathway [140].

Another common occurrence during oncogenesis is epithelial-to-mesenchymal transition, which has been shown to repress ULBP1 ligand expression in a gastric cancer model and further contribute to the immunosuppressive tumor microenvironment [141]. In additional studies related to cancer stressors, there is an increased expression of heat shock proteins, which facilitate the restoration of homeostatic signaling through refolding of denatured proteins and are known to be both upregulated under malignant conditions and regulate the expression of NKG2D ligands [142,143,144,145]. As illustrated above, increasing evidence suggests a relationship between NKG2D ligands and tumor initiation/maintenance. However, the impact of carcinogenesis on NKG2D ligands also depends on the immune cell influenced by the ligand expression. A study by Strid et al., using the epidermis as a model, reported that in response to local Rae-1 induction, certain immune cell populations exuded differing effects on carcinogenesis [146]. For example, with Rae-1 expression, gamma delta cells were able to protect the tissue from carcinogenesis, while Langerhans cells promoted it [146]. The immune cell responsive to NKG2D ligand induction could be one explanation for the diversity in cancer-mediated regulation of this pathway. Understanding the heterogeneous induction of this immune surveillance pathway is extremely important for targeting cancer cell vulnerabilities (Figure 2): however, there is also a necessity to understand compensatory mechanisms that cancer cells may employ to diminish pathway activity.

## 4. Biological Repression of NKG2D Ligands in Cancer

Mechanisms to combat NKG2D ligand-mediated recognition are commonly implored in cancer cells. Due to their high degree of plasticity and adaptability, cancer cells can manipulate their biological activity to evade the NKG2D immune surveillance pathway. Next, we highlight the established immunosuppressive mechanisms utilized by cancer cells to downregulate NKG2D ligands and avoid NKG2D-mediated killing (Figure 3).

### 4.1. Shedding

A regulatory mechanism of the immune surveillance pathway is the proteolytic cleavage of cell surface bound NKG2D ligands, which is also known as shedding. Two primary families of proteases associated with this shedding phenotype include matrix metalloproteinases (MMPs) and disintegrin and metalloproteinases (ADAMs). First established using epithelial-derived tumors, MICA was shown to be actively released by tumor cells as a means of immunosuppression [147]. Inhibition of metalloproteinases reversed the shedding phenotype and promoted the accumulation of MICA on the tumor cell surface [147]. NKG2D ligand shedding occurs in many other tumor types and has been validated in analyses from cancer patient serum [148,149,150,151,152]. Further supporting the complexity of the immune landscape, immune regulatory cytokines also have multi-faceted roles in NKG2D ligand expression, which is mediated partly by MMP cleavage. For example, interferon alpha (IFN-𝛼) has been shown to enhance MICA expression on tumor cells through increased promoter activity. Alternatively, interferon-gamma has been shown to enhance the cleavage of MICA via MMP9 [153]. As mentioned earlier, ADAM proteases are also responsible for the proteolytic shedding of NKG2D ligands. Using a standard model of glioblastoma, ADAM17 was shown to promote MICB solubilization through specific active sites at detergent-resistant membrane microdomains (DRMs) [154]. In addition to ADAM17, ADAM10 and ADAM9 have both been implicated in the shedding of NKG2D ligands [155,156,157,158]. Of note, hypoxia is a potent negative regular of NKG2D ligand expression: hypoxia directly influences both exosome release and the shedding of ligands [159]. Additionally, hypoxia diminishes the effector functionality of the NKG2D receptor expressing immune populations [160,161,162]. Relevant to clinical implications, investigators have sought to target hypoxia-induced immunosuppression in the hopes of enhancing immunotherapies [159,163,164,165,166,167].

### 4.2. Exosomes

One mechanism cancer cells use to regulate NKG2D ligands is their extracellular release through vesicular transport via exosomes [168]. In response to different stimuli, exosomes are intracellularly produced during endo-lysosomal processing and then released into the external microenvironment. Multiple tumor models including ovarian, mesothelioma, and cervical cancer have displayed the regulation of ligands through enhanced exosome production and release [169,170,171]. Specifically in leukemia cells, tumor-specific stressors, such as thermal and oxidative stress, promoted the release of exosomes containing NKG2D ligands that subsequently diminished NK cell cytotoxicity [172]. This exosome-mediated immunosuppressive phenotype has also been observed in prostate cancer cells where cancer cell-derived exosomes containing MICA/B, ULBP1, or ULBP2 led to the downregulation of NKG2D receptor expression on effector populations [173]. Elegantly proving this mechanism, the supplementation of tumor-derived exosomes to healthy lymphocytes led to the subsequent downregulation of NKG2D receptor expression [173]. In the broader context of the tumor microenvironment, exosomes derived from other cell populations, such as dendritic cells, regulate immune reactivity. Interestingly, dendritic cell-derived exosomes containing NKG2D ligands were able to enhance the proliferation and activity of NK cells through an IL-15Ralpha-mediated mechanism [174]. The exact mechanisms that regulate ligand secretion via exosomes remains to be elucidated, as exosomes are an evolving area of study.

### 4.3. Immunosuppressive Signaling Regulators

Many signaling pathways are aberrantly active under malignant conditions, and some of these may repress the expression and activity of NKG2D pathway intermediates. For example, transforming growth factor beta (TGF-β) has a multi-faceted role in cancer progression, where its effects on the tumor microenvironment can dampen immune cell activity [175]. In a lung cancer model, Lee et al. recently determined that TGF- β negatively regulated NGK2D ligand expression through the upregulation of MMP activity [176]. Similarly, TGF-β-suppressed NKG2D ligands MICA, ULBP2, and ULBP4 via MMP activity in a glioma model [177]. Importantly, in addition to repressing ligand expression, TGF- β has also diminishes the effector activity of NKG2D-bearing T-cells [178,179,180].

Another pathway implicated in the negative regulation of NKG2D ligand expression is the Janus Kinase and Signal Transducer and Activator of Transcription (JAK/STAT) pathway. In melanoma, MICA expression was negatively regulated by the production of interferon-gamma mediated by the activity of STAT1 [181]. In a castration-resistant pancreatic cancer model, the JAK/STAT signaling axis also regulated NKG2D ligand levels [182]. Specifically, IL-6 mediated the activation of JAK-STAT3, which led to a repression of multiple NKG2D ligands (MICA, MICB, ULBP1, ULBP2, and ULBP3) and evasion from NK cell killing [182]. Interestingly, this study also determined that the IL-6/JAK/STAT3 signaling cascade regulated the expression of the immune cell inhibitory molecule Programmed Death Receptor Ligand 1 (PD-L1), which is known inhibit effector cell activity [182,183,184]. Other studies have also demonstrated the negative regulation of NKG2D ligand expression via the PDL1/2 axis. For example, Okita et al. determined that MICA, ULBP2, ULBP5, and ULBP6 expression was induced in response to cisplatin [185]. However, along with this increase in NKG2D ligands, there was also an induction of PDL-1, diminishing the potential of NK cell-mediated killing [185]. A recent study by Lee et al. suggested that radiotherapy could induce expression of MICA, MICB, ULBP1, ULBP2, and ULBP3 in melanoma cells, but the concurrent induction of PD-L1 on the cells diminished the potential cytotoxic effect of the NK-92 cell line [186].

There is also increasing evidence that the epigenetic landscape regulates the expression patterns of NKG2D ligands. For example, histone deacetylases suppress NKG2D ligand expression. In epithelial tumor cells, both pharmacologic and genetic inhibition of histone deacetylase 3 (HDAC3) were able to rescue the suppressed expression of ULBP1-3. Mechanistically, it was determined that in response to stress, HDAC3 was recruited to the promoter regions of ULBP1-3, where it mediated the direct repression of these ligands. Recently, Histone Deacetylase 8 (HDAC8) was shown to modulate glioma immune responses through regulation of NKG2D ligands. Following pharmacologic inhibition of HDAC8, chromatin immunoprecipitation sequencing (ChIP) analysis identified a direct regulation of MULT-1, Rae-1, and H60 murine NKG2D ligands [187]. Collectively, the diverse regulatory mechanisms for NKG2D ligand expression provides promising therapeutic targets.

## 5. Therapeutic Approaches Utilizing NKG2D Ligand Expression

Identifying treatments that induce NKG2D ligands to enhance immune cell-mediated killing is a promising immunotherapy approach (Table 1). While they have off-target effects, standard of care radio- and chemotherapy can alter the NKG2D pathway [188,189,190]. In glioblastoma, radiation and the DNA alkylating agent, temozolomide, were shown to induce MICA, MICB, ULBP2, and ULBP3 ligand expression in vitro and in vivo, which conferred higher immunoreactivity [191]. Additionally, temozolomide was able to induce multiple NKG2D ligands (MICA/B, ULBP1-3) in a glioblastoma stem cell model [106]. These findings support previous work performed by Lamb et al. where a transient induction of MICA/B, ULBP1-ULBP3 ligands was observed in standard models of glioblastoma after treatment with temozolomide [93]. Interestingly, the addition of gamma delta T-cells engineered to be temozolomide-resistant (through overexpression of the DNA repair enzyme methylguanine DNA methyltransferase (MGMT)) to temozolomide treatment increased overall survival in mice bearing glioblastoma xenografts [192,193]. Based on these results, temozolomide-resistant gamma delta T cells combined with standard of care temozolomide are being investigated in a phase 1 clinical trial as upfront maintenance therapy in adults with glioblastoma (NCT04165941).

The heterogeneous regulation of NKG2D ligands provides multiple potential avenues for immunotherapies that enhance ligand expression. As illustrated in Figure 2, TGF- β is a negative regulator of NKG2D ligand expression. Either genetic or pharmacologic inhibition of this pathway has shown promise at restoring ligand expression and effector cell activity [194,195]. Recently, in a preclinical breast cancer model, Liu et al. observed therapeutic efficacy of an encapsulated combination of TGF-β inhibitors with selenocystine. The combination enhanced NK cell-mediated tumor cell death through the induction of MICA, MICB and ULBP1-4 [196]. In addition, HDAC inhibitors are a promising approach and have proven successful for enhancing NKG2D ligand expression in a variety of cancer models [197,198,199,200,201,202]. Another approach being explored is inhibiting MMP activity to decrease the shedding activity that may limit cytotoxic effects [136,137,138]. In addition to targeting the negative regulators of NKG2D ligands, activating the positive regulators of NKG2D ligands has proven successful in a variety of cancer types. For example, DDR activation through small molecule inhibition can increase NKG2D ligands. Furthermore, heat shock protein inhibitors (i.e., HSP90), inhibitors of apoptosis (IAP), and proteosome inhibitors have been shown to enhance the expression of a variety of NKG2D ligands in different cancer types via the activation of DDR pathway mediators [203,204,205].

Another promising cancer therapeutic that may be responsive to NKG2D pathway manipulation is oncolytic virotherapy [206,207,208]. As a well-established stressor to induce NKG2D ligand expression, viral infection targeting malignant cells may result in ligand induction to potentiate the cytotoxic effect of NKG2D bearing immune populations [209,210,211,212,213]. As cancer cells are highly adaptable, there are mechanisms to evade viral-induced NKG2D ligand expression. Therefore, strategic manipulation is necessary to maximize therapeutic effect. Table 1 summarizes current clinical trials utilizing NKG2D pathway manipulation (Table 1). As a by-product of ligand induction, an added advantage may be the identification of predictive biomarkers to support NKG2D-driven immunotherapy [214,215,216,217].

Although this notion of NKG2D ligand induction for immunotherapies is exciting, the heterogeneity of tumors across tissue types, and even the intratumoral heterogeneity amongst patients, makes universal prognostic indicators extremely difficult. In certain instances, there is a benefit to having NKG2D ligands robustly induced and expressed. However, in other tumors, sustained NKG2D ligand overexpression is a negative indicator of overall patient survival. For example, from an immunohistology screen, MICA/B and ULBP1 were shown to be both correlated and good prognostic indicators for cervical cancer patient survival [215]. On the other hand, a recent bioinformatic analysis of ULBP1 expression in colon adenocarcinoma patients revealed ULBP1 as a negative indicator of overall patient survival [218]. Interestingly, in colorectal cancer, MICA and ULBP5 (RAET1G) co-expression was shown to positively influence overall patient survival [219]. However, their expression was most abundant in stage 1 metastatic tumor nodes, while expression decreased as the tumor stage increased [219]. The tissue and patient-specific reasoning behind differential survival responses to NKG2D ligand expression has not been clearly elucidated, and further research investigating the regulation of this pathway that carefully considers heterogeneity, including in preclinical models, is needed.

## 6. Conclusions

Employing the NKG2D stress response pathway for therapeutic benefit is an exciting immunotherapy strategy. While significant strides have been made in the foundational understanding of these immune-modulating ligands, there is still ample room to increase our understanding to utilize this pathway more precisely for clinical benefit. The diverse regulation of NKG2D ligands provides numerous therapeutic opportunities. There remains a need for further investigation of the mechanistic role that oxidative stress has on NKG2D ligand expression. In addition, a better understanding of the tumor microenvironment is critical. The nutrient availability, oxygen tension, and overall acidity of the tumor microenvironment largely affect signaling cascades. The large-scale implications of these environmental conditions have not been well established in the context of NKG2D ligand expression and may provide needed opportunities to enhance the effectiveness of immunotherapies. Additionally, further investigation into tumor-specific niches and regulation is necessary.

As evidenced by the diverse clinical responses amongst different tumor types, variability in NKG2D ligand-based immunotherapy efficiency is apparent: this warrants further investigation to identify strategies that may need to be disease-specific to overcome this differential response. For example, blood-borne (liquid) malignancies have often been utilized as models for the foundational mechanistic studies for NKG2D ligand biology, and these tumor types have shown significant success this specific type of immunotherapy [109,119,152,220]. However, due to the cellular heterogeneity and physiologic complexity that many solid tumors possess, targeting solid tumors with NKG2D ligand induction may have more therapeutic hurdles. Resulting from the diverse nature of solid tumors, the expression of NKG2D ligands can also be heterogeneously induced depending on the stimuli. As many cells within the tumor microenvironment differ in both therapeutic sensitivity and antigen generation/presentation, there is a benefit to utilizing stress-responsive NKG2D ligands. Under stressful stimuli, multiple ligands may be induced on differing cell populations within the tumor. These stimuli may arise from a variety of sources such as the different chemotherapeutic agents commonly employed as first-line treatments to newly diagnosed cancers. Despite the potential NKG2D ligand induction being diverse, with ligands displaying varying induction time/stability, intratumoral heterogeneity, and abundance, there is still the potential therapeutic benefit as the tumor becomes more recognized by the immune system. This enhanced recognition by effector cells may allow for the better implementation of cell-based immunotherapies by mechanistically making historically immunogenically cold tumors (inactive) turn hot (active). To illustrate this point further, some chemotherapies that have been investigated for NKG2D ligand induction in cancer are summarized below (Table 2). In summation, targeting the NKG2D stress response pathway has shown extraordinary promise in preclinical settings for treatment of cancer, and with further research, there is an opportunity to implement these strategies to maximize therapeutic efficacy.

## Figures and Tables

**Figure 1 cancers-14-02339-f001:**
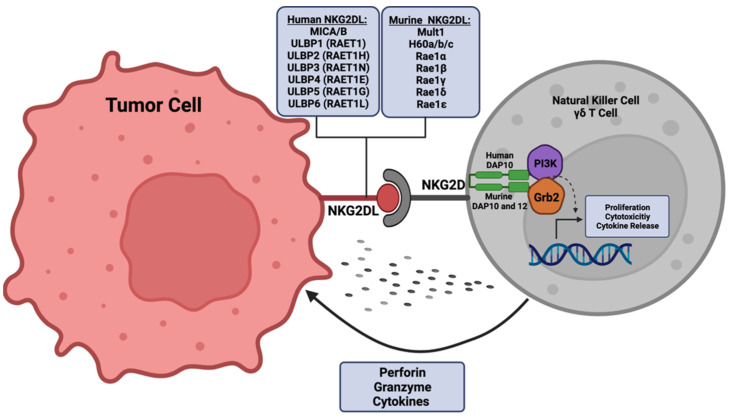
NKG2D ligand (NKG2DL) binding activates the cytotoxic potential of effector cells. In response to stressors, human or murine NKG2D ligands are expressed on the surface of target cells. NKG2D ligands bind to the respective NKG2D receptor found on cells such as natural killer (NK) cells or gamma-delta T cells. Utilizing adaptor molecules DAP10 or DAP12 (murine), the signal transducers, PI3K and Grb2, are recruited to the NKG2D-binding domain. Through intracellular signaling cascades, transcriptional activation of target genes associated with effector cell maintenance, including those regulating proliferation, cytotoxicity, and cytokine release, is initiated. Image generated using Biorender.com (accessed on 22 April 2022), licensure certificate available as Appendix A.

**Figure 2 cancers-14-02339-f002:**
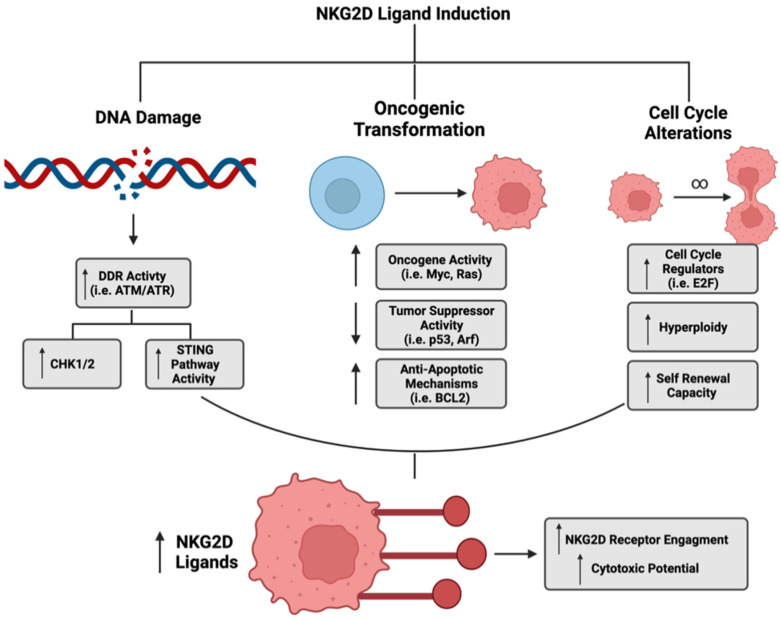
Mechanisms associated with NKG2D ligand induction. Commonly activated pathways associated with cancer progression may influence stress ligand expression. The DNA damage response pathway, which is activated during cancer treatments, increases NKG2D ligand expression via activation of signaling kinases ataxia telangiectasia mutated (ATM) and ataxia telangiectasia and RAD3 related (ATR). ATM/ATR promote the activation of downstream targets that increase and maintain ligand expression. Genetic alterations, such as the oncogenic hits that promote tumorigenesis and behavioral phenotypes, including excessive proliferation and “epithelial-to-mesenchymal” transition, associated with tumor biology are also important regulators of NKG2D ligand expression. Image generated using Biorender.com (accessed on 16 March 2022), licensure certificate available as Appendix A.

**Figure 3 cancers-14-02339-f003:**
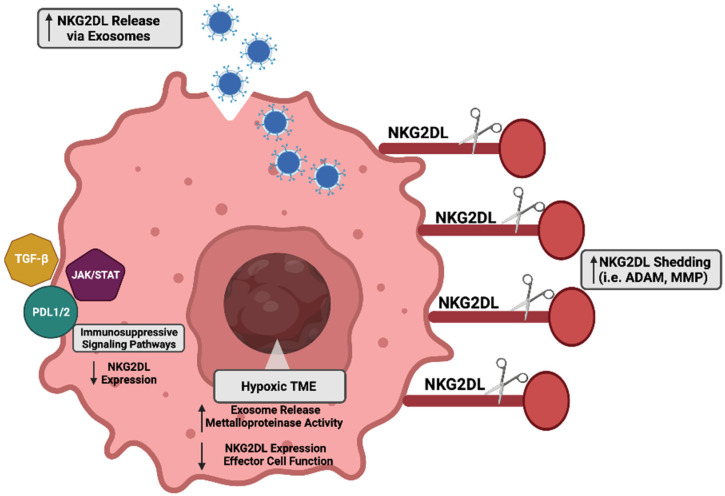
Repression mechanisms of NKG2D ligands in cancer. Multiple mechanisms have been identified in cancer cells to evade immune cell recognition. Generation and release of NKG2D ligands into the extracellular space via exosomes is enhanced and stimulated by the hypoxic tumor microenvironment. Similarly, enhanced proteolytic cleavage activity of matrix metalloproteinase (MMP) and a disintegrin and metalloproteinase (ADAM) family members is regulated by the hypoxic tumor microenvironment. Immunosuppressive and inhibitory mediators (Transforming Growth Factor Beta (TGF-β), Janus Kinase and Signal Transducer and Activator of Transcription (JAK/STAT), and Programmed Death Receptor Ligand 1 and 2 (PDL1/2)) also collectively diminish NKG2D ligand expression by potentiating the immunosuppressive landscape in cancer. Image generated using Biorender.com (accessed on 16 March 2022), licensure certificate available as Appendix A.

**Table 1 cancers-14-02339-t001:** Current clinical trials utilizing the NKG2D stress response pathway.

Study Title	Disease Targeted	Therapeutic Approach	Clinical Trials Identifier
NKG2D-based CAR T-cells Immunotherapy for Patient with r/r NKG2DL+ Solid Tumors	Hepatocellular Carcinoma, Glioblastoma, Medulloblastoma, Colon Cancer	Autologous genetically modified anti-NKG2DLs CAR transduced T cells	NCT05131763
NKG2D CAR-T Cell Therapy for Patients With Relapsed and/or Refractory Acute Myeloid Leukemia	Acute Myeloid Leukemia	NKG2D CAR T cells	NCT04658004
Pilot Study of NKG2D CAR-T in Treating Patients with Recurrent Glioblastoma	Recurrent Glioblastoma	NKG2D CAR T cells	NCT04717999
alloSHRINK—Standard cHemotherapy Regimen and Immunotherapy with Allogeneic NKG2D-based CYAD-101 Chimeric Antigen Receptor T-cells	Unresectable Metastatic Colorectal Carcinoma	Allogeneic NKG2D-based CYAD-101 Chimeric antigen Receptor T-cells, 5-FU, leucovorin, oxaliplatin, irinotecan	NCT03692429
Safety Study of Chimeric Antigen Receptor Modified T-cells Targeting NKG2D-Ligands	Acute Myeloid Leukemia, Multiple Myeloma Myelodysplastic Syndrome	NKG2D CAR T cells	NCT02203825
Pilot Study of NKG2D-Ligand Targeted CAR-NK Cells in Patients With Metastatic Solid Tumours	Solid Tumors	CAR-NK Cells followed by IL-2 injection	NCT03415100
Immunotherapy of CD8+ NKG2D+ AKT Cell With Chemotherapy to Pancreatic Cancer	Pancreatic Ductal Adenocarcinoma	CD8+, NKG2D+, AKT cells, Gemcitabine	NCT02929797
NKG2D CAR-T(KD-025) in the Treatment of Relapsed or Refractory NKG2DL+ Tumors	Solid Tumor, Hepatocellular Carcinoma, Colorectal Carcinoma, Glioma	Autologous genetically modified anti-NKG2DLs CAR transduced T cells	NCT04550663
Adoptive Cellular Immunotherapy Following Autologous Peripheral Blood Stem Cell Transplantation for Multiple Myeloma	Myeloma, Transplant Eligible Patients	Cytotoxic T-cells, IL-2, GM-CSF	NCT00439465
Novel Gamma-Delta (γδ) T Cell Therapy for Treatment of Patients with Newly Diagnosed Glioblastoma (DRI)	Glioblastoma	Gene modified drug resistant immunotherapy (γδT Cell) administered	NCT04165941

**Table 2 cancers-14-02339-t002:** Chemotherapies investigated in cancer for NKG2D ligand induction.

Chemotherapy	Cancer Type	Ligands Induced	Reference(s)
Cisplatin	Lung	MICA/B; ULB2/5/6	[185,221]
Bortezomib	Multiple Myeloma AML, ALL	MICA/B; ULBP1/2/3/5/6	[222,223,224]
Gemcitabine	Lung, Hepatocellular, Colorectal	MICA/B; ULBP1/2/3/5/6	[225,226,227]
5-Fluorouracil	Pancreatic Cancer, Lung	MICA/B; ULBP1/2/4/5/6	[228,229]
Pemetrexed	Lung	MICA/B; ULBP2/5/6	[230]
Vemurafenib	Melanoma	MICA; ULBP2	[231]
Decitabine	Osteosarcoma, IDH Mutant Glioma	MICB; ULBP1/3	[105,232,233]
Temozolomide	Glioblastoma	MICA/MICB; ULBP1/2/3/4	[93,191,192]
Metformin	Leukemia	ULBP1	[234]
Gefitinib	Lung	MICA; ULBP1/2	[226,235]
Erlotinib	Lung	MICB; ULBP1	[236]
Dacarbazine	Melanoma	Rae-1; Mult-1	[237,238]
Sunitinib	Nasopharyngeal	MICA/B; ULBP1/2/3	[239,240]
Trabectedin	Multiple Myeloma	MICA/B; ULBP1	[241]
Sulforaphane	Breast, Adenocarcinoma, Lymphoma	MICA/B	[242]

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
