# Peer review of "Regulation of NKG2D Stress Ligands and Its Relevance in Cancer Progression"

_cancers, 2022, doi:10.3390/cancers14092339_

Round 1

Reviewer 1 Report

The work is well written and provide a broad review about NKG2D mediated mechanism related with cancer with relevant references, only minor aspect should be considered:

-There is data in the literature that describe genetic variations in NKG2D gene (polymorphisms) associated with various pathologies including cancer and relate with the efficacy of NKG2D-mediated antitumor response and clinical outcome, that are not discussed in the text (e.g. Hayashi et al., 2006 or Gimeno et al., 2019)

-some errors (minor):

  • line 312: there is a typing mistake in IFN type molecule
  • Table 1 (current clinical trials): should be review since the last rows are mismatched

Reviewer 2 Report

In this manuscript, Amber and Dr. Hjelmeland prepared a review article summarizing the current understanding of regulatory mechanisms controlling expression of NKG2D ligands in cancer. Overall, it is an interesting and update review manuscript; However, the novelty and strengths of this manuscript are limited, since there are already many publications reported that potential roles of NKG2D and its ligands in cancer immunotherapy. Meanwhile, the tissue specific and temporal manner of NKG2D ligands expression in human development should be discussed in this review, and the diagnostic and prognostic values of NKG2D and its ligands in cancer should be also included in this review.

Reviewer 3 Report

  1. Various kinds of stresses upregulated the NKG2D ligands during tumorigenesis. Indeed, there are numerous NKG2D ligand-upregulated cases in different types of tumors (carcinomas of the breast, lung, colon, ovary, kidney, and prostate, melanomas, gliomas, and leukemia). Tumors induce NKG2D ligand suppression because of their high-upregulated population. In the introduction, there is no mention of tumors overexpressing NKG2DL, and we could be confused that NKG2D ligand expression is decreased under the average, resulting from NKG2DL dysregulation. Therefore, it would be better to ensure the step in which cancer cells attempt to escape immune surveillance through various regulatory mechanisms in specific condition after increasing NKG2D ligands during tumorigenesis.
  2. Chemotherapy makes differences in tumor cells after treatment. Sometimes, they suppress the NKG2DL expression, but sometimes they induce it. It seems pretty different by cell/tumor types or types of ligands. So it is hard to say that chemotherapy downregulates NKG2DL. Explaining these heterogeneous results or focusing on a specific tumor/ligand seems much better. 

Round 2

Reviewer 2 Report

The authors have satisfactorily responded to  all my comments, and made necessary changes to the manuscript.